# The impact of insect herbivory on biogeochemical cycling in broadleaved forests varies with temperature

Herbivorous insects alter biogeochemical cycling within forests, but the magnitude of these impacts, their global variation, and drivers of this variation remain poorly understood. To address this knowledge gap and help improve biogeochemical models, we established a global network of 74 plots within 40 mature, undisturbed broadleaved forests. We analyzed freshly senesced and green leaves for carbon, nitrogen, phosphorus and silica concentrations, foliar production and herbivory, and stand-level nutrient fluxes. We show more nutrient release by insect herbivores at non-outbreak levels in tropical forests than temperate and boreal forests, that these fluxes increase strongly with mean annual temperature, and that they exceed atmospheric deposition inputs in some localities. Thus, background levels of insect herbivory are sufficiently large to both alter ecosystem element cycling and influence terrestrial carbon cycling. Further, climate can affect interactions between natural populations of plants and herbivores with important consequences for global biogeochemical cycles across broadleaved forests.

Herbivory is an important mediator of ecosystem nutrient cycling and primary production across biome types[1,2]. A wide diversity of herbivores shape the form, function, and biochemistry of plants, exhibiting deep and taxonomically diverse co-evolutionary linkages to plants[3]. The impacts of mammalian herbivores and those of a small group of insects that cause extensive but rare mass defoliation events have received significant attention[4,5]. However, a more cryptic, diverse, and extensive community of insects is responsible for near-continuous and ubiquitous background levels of herbivory. The seemingly minor contributions of background insect herbivory to ecosystem processes under non-outbreak conditions may be substantial over the long term and over large spatial scales, with ecosystem consequences that likely differ from the more charismatic yet sporadic outbreak events[6]. The magnitude of these impacts, the variation of these impacts across the world's forests, and importantly for terrestrial ecosystem modeling, the drivers of this variation all remain poorly quantified[6,7].

An expanded focus on insect herbivores is also warranted because they create important feedbacks between plants and soils mediated by a wide variety of mechanisms[8]. One key direct, immediate feedback occurs via transfer of labile nutrients from green leaves to the soil in the form of excreta, cadavers, leachate, unconsumed leaf fragments, and prematurely abscised leaves (Fig. 1)[8,9]. Relative to leaf litter, herbivory-related insect deposits are typically enriched with labile forms of nutrients, and in many forests, insect-mediated nutrient fluxes are comparable to or even exceed fluxes from other inputs of relatively labile, mineral forms of nutrients[9,10]. In contrast, most of the nutrients in leaf litter and are resorbed and retained within plant biomass, or they are released in relatively recalcitrant forms[11]. When folivores alter the fluxes of limiting nutrients such as nitrogen (N) and phosphorus (P), they also have the potential to influence plant growth and ecosystem carbon (C) cycling[11,12]. Silicon (Si) is increasingly investigated in plant science research because silica enhances plant structural integrity, reduces the impact of stressors such as herbivory and drought, and correlates with C sequestration[13,14]. However, the biogeochemical dimensions of herbivory impacts on Si remain understudied[15]. Though global analyses of herbivory exist e.g.[16], the flux of nutrients associated with insect herbivory and insect deposits remains poorly understood[8], as is the potential impact of climate on these fluxes.

✉e-mail: bernice.hwang@uibk.ac.at

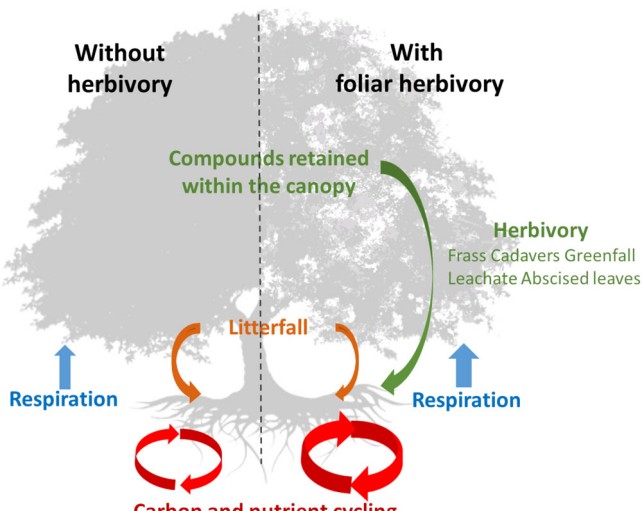

**Fig. 1 | Hypothesized effects of insect herbivory on ecosystem element cycling in a broadleaved forest.** Although herbivores exert a wide variety of other direct and indirect effects, our study focused on one major direct effect of herbivores—removal of foliar matter. Briefly, insect foliar herbivory (H) constitutes an important pathway for labile carbon and nutrients to move from green foliage to the soil—via excreta, cadavers, unconsumed leaf fragments, early abscised leaves, and leachate. Once these products of H enter soil, contained nutrients alter a range of processes that support soil microbial communities and plant growth. Foliar production (FP), while not tested here, would be negatively affected by H at the individual target plant level, but neutrally or positively affected by H at the stand level as adjacent non-target trees, composed of herbivore resistant genotypes or species, benefit from access to additional fluxes of growth limiting nutrients. Similarly, greenleaf nutrients ($F_E$) would decline at the target plant level as a result of H, but for similar reasons as FP, would remain unchanged or even increase at the stand level. $RE_E$ represents the difference in element content between green and freshly senescent leaves, the quantity of which would be absorbed by the tree prior to senescence. $H_c$ (FP x $F_E$ x H) represents the gross amounts of elements consumed by insect folivores, and $H_i$ ($L_{EH} + H_c - L_F$) refers to the additional (net) element inputs from insect folivores due to release of nutrient rich green leaf material prior to resorption[10]. In all cases, the subscript E refers to elements. Arrow sizes denote the relative size of the flux. Herbivory-related calculations are fully described in Supplementary Table 1. Tree silhouette adapted from NikhomTreeVector/Shutterstock.com.

To test fundamental hypotheses about the magnitude of nutrient fluxes mediated by background levels of insect herbivory, the variation of these fluxes across the Earth's angiosperm forests, and the global-scale drivers of this variation, we established 74 plots within 40 mature, undisturbed broadleaved forests representing nearly the full range of broadleaved forests on Earth[17]. We used standardized methods to regularly collect and analyze green and freshly senesced leaves throughout the growing season for one or two years at each plot for: (*i*) foliar biomass production, (*ii*) foliar herbivory, and (*iii*) foliar C, N, P and Si concentrations. We then used these measures to calculate annual stand-level element fluxes generated during leaf consumption by entire, natural communities of insect herbivores under non-outbreak conditions[10]. We later compared these fluxes to other sources of labile nutrients (atmospheric N, atmospheric P, and bedrock weathered P). Furthermore, we investigated potential abiotic and biotic drivers of these insect-mediated C and nutrient fluxes. High nutrient availability, warm temperatures, and low water stress can positively influence foliar nutrient concentrations, foliar biomass production, and insect abundance with positive synergistic overall effects on insect-mediated element fluxes[18–20]. Consequently, warm mean annual temperatures (MAT) and high soil nutrient concentrations could increase insect herbivory and thus insect-mediated element fluxes in nutrient-limited but not water-limited (i.e., low dryness) systems. Therefore, we hypothesized that:

H1—The flux of N and P from plant to soil mediated by insect herbivores would meet or exceed other major labile fluxes of these elements, such as atmospheric deposition (for N and P) and bedrock weathering (for P).

H2—Insect-mediated element fluxes would increase with increasing mean annual temperature, decrease with increasing dryness (potential evapotranspiration/mean annual precipitation), and increase with increasing soil nutrient concentration.

H3—Observed responses of insect mediated element fluxes to MAT, dryness and soil nutrient concentrations would be driven in equal measure by similar responses from foliar herbivory, biomass production and foliar element concentrations.

Here, we show that background levels of insect herbivory can have profound impacts on biogeochemical cycling in broadleaved forests. For some localities, these fluxes exceed atmospheric deposition inputs, showing that background levels of insect herbivory are large enough to both alter ecosystem element cycling and, because primary productivity in most forests is nutrient-limited, influence terrestrial C cycling. Insect-mediated fluxes of N and P are especially high in tropical forests compared to temperate and boreal forests. Further, we show that all insect-mediated element fluxes increase strongly with MAT. These results reveal how climate can affect interactions between natural populations of plants and herbivores with important consequences for C and nutrient cycling across global broadleaved forest biomes.

## Results and discussion
### Forest characteristics across our global network of research plots

MAT across our global network of plots ranged from −1.4 – 26.9 °C (Supplementary Table 2), which represents nearly the full range of mean annual temperature (MAT) for broadleaved trees on Earth. Similarly, dryness, represented by potential evapotranspiration/mean annual precipitation, ranged from 0.21 – 1.30, capturing a very broad range of moisture conditions for broadleaved forests (Supplementary Table 2). Mean soil element concentrations were 0.8–27.2% for C, 0.04–1.81% for N, 0.0004–0.2785% for P, and 0.1–24.7% for Si (Supplementary Table 2). Capturing most of the global range for foliar biomass production in broadleaved forests (Table 1, Supplementary Table 3, and Supplementary Fig. 2), the mean for tropical forests (539.3 ± 42.6 g m$^{-2}$ y$^{-1}$) was 66% larger than temperate foliar biomass production (323.4 ± 28.6 g m$^{-2}$ y$^{-1}$) and 240% greater that boreal forest foliar biomass production (158.8 ± 47.6 g m$^{-2}$ y$^{-1}$). We found that foliar herbivory (henceforth herbivory) was widespread, with an average of 48.8 ± 2.4% of all leaves globally showing some level of damage from folivores. Mean herbivory rate per leaf was 4.1 ± 0.1% globally, with herbivory rate across plots ranging from 0.6 ± 0.1% – 13.6 ± 0.8% (Supplementary Table 2). Herbivory showed a strong gradient across latitude zones, ranging from a high of 5.4 ± 0.2% in tropical forests, to 3.3 ± 0.1% in temperate forests, to the low of 2.5 ± 0.2% in boreal forests (Table 1, Supplementary Table 3 and Supplementary Fig. 2). Resorption efficiency for N was significantly greater in boreal (61.8 ± 3.3%) than in tropical (46.5 ± 2.7%) forests while resorption efficiency for P was significantly greater in tropical (66.5 ± 2.1%) than in temperate (57.9 ± 2.1%) or boreal (55.2 ± 4.5%) forests (Supplementary Table 3 and Supplementary Fig. 2). Resorption efficiency for C globally was 19.7 ± 0.4% (Supplementary Table 2).

### Insect-mediated fluxes compared to other major sources of labile nutrients

Net insect-mediated element fluxes ($H_i$) exceeded that of some other labile element sources in some locations. The global mean for $H_i$C was 1.7 ± 0.1 g m$^{-2}$ y$^{-1}$, 0.17 ± 0.02 g m$^{-2}$ y$^{-1}$ for $H_i$N, and 0.014 ± 0.002 g m$^{-2}$ y$^{-1}$ for $H_i$P (Supplementary Table 2). $H_i$N and $H_i$P were greatest at tropical sites, averaging 68% of N deposited atmospherically, 440% of P

**Table 1 | Gross insect-mediated element fluxes in broadleaved forests across latitude zones**

| Variable | Units | Tropical | Temperate | Boreal | $\chi^2$ | DF | P |
|---|---|---|---|---|---|---|---|
| Foliar biomass production (FP) | g m$^{-2}$ y$^{-1}$ | 513.0 ± 10.2$^a$ | 321.6 ± 7.3$^b$ | 231.9 ± 17.5$^c$ | 32.05 | 2 | <0.001 |
| Foliar herbivory (H) | % leaf area removed | 5.4 ± 0.2$^a$ | 3.3 ± 0.1$^b$ | 2.4 ± 0.2$^b$ | 8.49 | 2 | 0.014 |
| Gross insect-mediated C flux (H$_c$C) | g m$^{-2}$ y$^{-1}$ | 15.87 ± 0.90$^a$ | 5.41 ± 0.90$^b$ | 1.65 ± 0.58$^c$ | 24.37 | 2 | <0.001 |
| Gross insect-mediated N flux (H$_c$N) | g m$^{-2}$ y$^{-1}$ | 0.600 ± 0.038$^a$ | 0.231 ± 0.038$^b$ | 0.082 ± 0.029$^b$ | 14.21 | 2 | <0.001 |
| Gross insect-mediated P flux (H$_c$P) | g m$^{-2}$ y$^{-1}$ | 0.034 ± 0.003$^a$ | 0.016 ± 0.003$^b$ | 0.006 ± 0.002$^b$ | 12.17 | 2 | 0.002 |
| Gross insect-mediated Si flux (H$_c$Si) | g m$^{-2}$ y$^{-1}$ | 0.178 ± 0.076$^a$ | 0.178 ± 0.76$^{ab}$ | 0.029 ± 0.012$^b$ | 7.04 | 2 | 0.030 |

Mean foliar biomass production, foliar herbivory, and gross insect-mediated element fluxes across tropical (33), temperate (32), and boreal (9) forest plots with standard errors and associated descriptive statistics, chi-square ($\chi^2$), degrees of freedom (DF), and P values (P). Different letters following means indicate significant differences based on two-sided Kruskal-Wallis and Dunn-Bonferroni post-hoc tests (95% CI). Source data are provided as a Source Data file.

**Table 2 | Comparison of nitrogen and phosphorus sources in broadleaved forests across latitude zones**

| Latitude zone | H$_i$N | H$_i$P | Atmospheric N* | Atmospheric P* | Bedrock weathered P** |
|---|---|---|---|---|---|
| Tropical | 0.26 ± 0.04 | 0.022 ± 0.004 | 0.38 ± 0.08 | 0.005 ± 0.001 | 0.054 ± 0.019 |
| Temperate | 0.11 ± 0.02 | 0.009 ± 0.002 | 1.03 ± 0.14 | 0.004 ± 0.001 | 0.045 ± 0.015 |
| Boreal | 0.05 ± 0.02 | 0.003 ± 0.001 | 0.49 ± 0.12 | 0.003 ± 0.001 | 0.069 ± 0.037 |
| Global mean | 0.17 ± 0.02 | 0.014 ± 0.002 | 0.69 ± 0.08 | 0.005 ± 0.001 | 0.056 ± 0.012 |

* Estimates derived from model by Brahney et al.[54]
** Estimates derived from model by Hartmann et al.[55]

Net insect-mediated fluxes (H$_i$, g m$^{-2}$ y$^{-1}$) of nitrogen (N) and phosphorus (P) compared to other major ecosystem inputs of nutrients in 33 tropical, 32 temperate, and 9 boreal forest plots. Values represent means ± standard errors. Source data are provided as a Source Data file.

deposited atmospherically, and 41% of estimated mean bedrock weathered P for the region (Table 2). In contrast, H$_i$N and H$_i$P were lowest at boreal sites, averaging 10% of the N deposited atmospherically, matching mean atmospheric P deposition rates, and 4% of mean bedrock weathered P for the region (Table 2). H$_i$N was greater than the regional mean for atmospheric N deposition at 29% of the tropical plots but none of the temperate and boreal plots. H$_i$P exceeded the regional mean of atmospheric P deposition at 73% of the tropical plots, 69% of the temperate plots, and 33% of the boreal plots.

Background levels of insect herbivory represented a small but persistent pathway for the return of foliar organic material and elements from broadleaved forest canopies to the soil (Table 2, Supplementary Table 2). In partial agreement with Hypothesis 1, net insect-mediated nutrient fluxes from this pathway exceeded that of atmospheric nutrient deposition in some localities (Table 1). For some sites, H$_i$N and H$_i$P exceeded that of atmospheric N (21% of locations) and atmospheric P (72% of locations). Insect-mediated N and P fluxes could have particular importance in systems that are limited by the availability of these nutrients. For example, in the tropics where soil P availability can be very low[21,22], the relative importance of insect-mediated P could have greater implications for P cycling compared to other regions. That we found H$_i$P and soil C:P to be greatest in tropical forests (Table 2, Supplementary Table 2) underscores the potentially important role of insect-mediated P fluxes in this zone. Further, H$_c$P:Litter P was significantly higher in tropical forests than in either temperate or boreal forests. Resorption is a mechanism for addressing high soil P fixation, especially in acidic and heavily weathered soils[22]. Therefore, H$_i$P might be more easily fixed by soil (into non-available inorganic forms), which over time would drive down P availability. Meanwhile, productivity especially in cooler regions is generally projected to increase as are forest disturbances (e.g., drought, fire) because of climate change[23,24]. Therefore, greater foliar biomass production and herbivory due to temperature increases could amplify the importance of some insect-mediated element fluxes in cooler forests (Supplementary Table 6).

Our work aligns with previous research[10,25] showing that relatively small but continuous levels of background herbivory may affect long-term ecosystem C and nutrient cycling. These effects can be as large or larger than those driven by more visible and generally better studied episodic outbreaks[26,27]. That is, annual losses of foliar biomass due to background insect herbivory at regional or global scales could exceed the annual loss caused by local outbreaks of forest pests[6]. The timing of the release of those nutrients is also important to consider. During a typical spatially and temporally isolated insect outbreak, nutrient release occurs over a relatively short time period during which demand for nutrients may be reduced because of herbivore impacts on plant function. For elements such as P, a pulse of P availability could shift the ecosystem stock of P into less available forms, as abiotic fixation of P occurs rapidly, especially in the acidic soils of most forests[28]. Conversely, background herbivory may cause small but continuous inputs of P that are less susceptible to geochemical fixation by soils. Previous work has shown that in some systems, small but labile herbivory related inputs can relieve plant nutrient limitations[29,30].

Though our study focused on one major direct effect of herbivores—removal of foliar matter—herbivores exert a wide diversity of other effects, both direct and indirect. For example, changes in plant productivity due to compensatory growth or isoprene emissions can then lead to feedbacks affecting herbivore performance and activity[31,32]. Folivores can stimulate microbial biomass and activity via pulses in root exudation or encourage the growth of well-defended plants that produce recalcitrant litter and reduce soil activity[8]. We did not account for shifts in plant properties, neither in some of the biotic precursors included in this study, as they adapt to herbivory, nor through within-species plasticity or via shifts in plant community composition[8]. Other insect feeding guilds (e.g., root herbivores, sap suckers) are also likely to have different effects on ecosystem processes than folivores[31]. Future work on these and other direct and indirect effects are needed to further our understanding of insect herbivore impacts on element cycling.

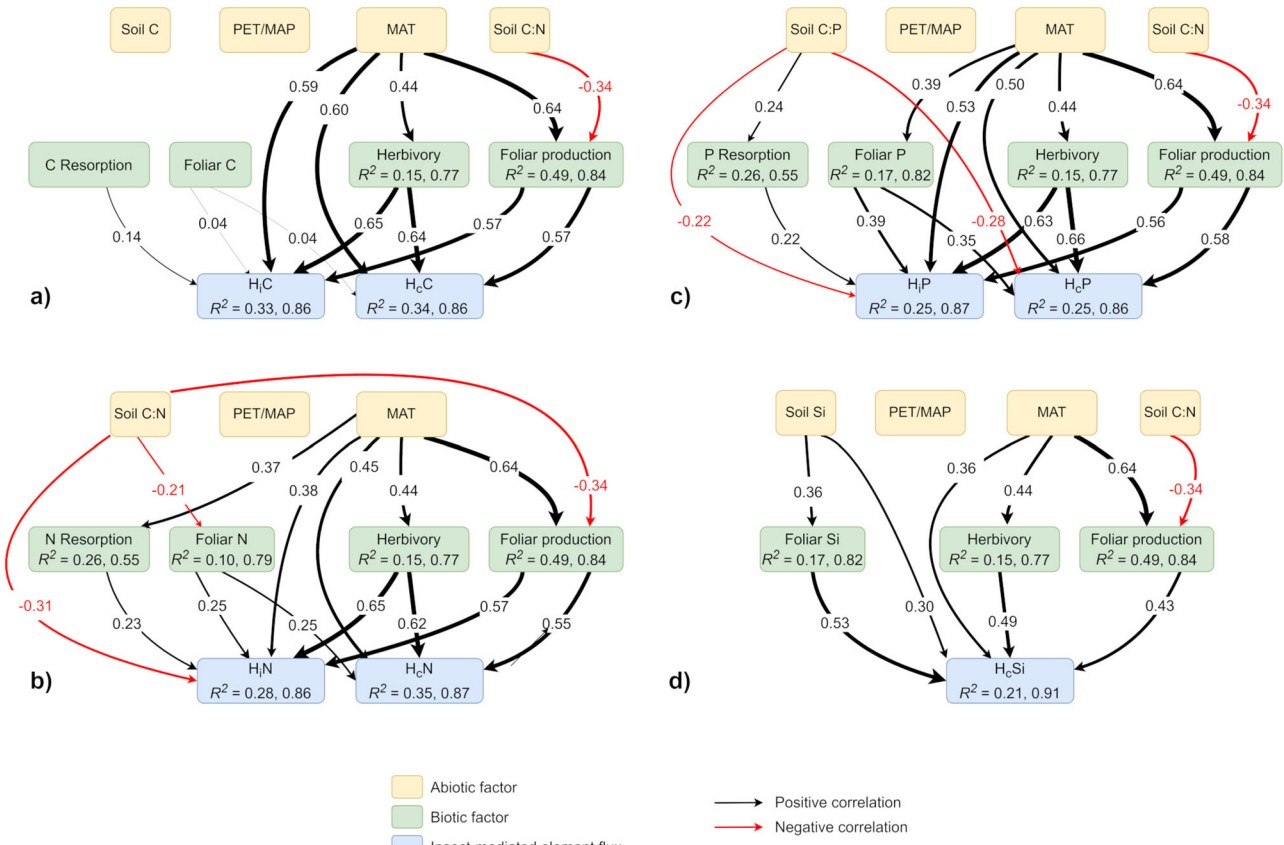

**Fig. 2 | Drivers of insect-mediated element fluxes in broadleaved forests.** Proposed pathways for the influence of abiotic variables (yellow nodes) and biotic precursors (green nodes) on global insect-mediated (**a**) carbon, (**b**) nitrogen, (**c**) phosphorus, and (**d**) silica gross (H$_c$, g m$^{-2}$ y$^{-1}$) and net (H$_i$, g m$^{-2}$ y$^{-1}$) fluxes (blue nodes). Abiotic variables in the full model included mean annual temperature (MAT, °C), dryness ratio expressed as potential evaporation/mean annual precipitation (PET/MAP), and soil element concentration (C, Si) or stoichiometries (C:N, C:P). Biotic precursors to insect-mediated fluxes in the full models included foliar biomass production (g m$^{-2}$ y$^{-1}$), foliar herbivory rate (% leaf area removed y$^{-1}$), foliar element concentration (%), and resorption efficiency (%). Some variable nodes are identified with the $R^2$ value (marginal, conditional) from multiple regression represented by incoming arrows originating from abiotic variables. Black arrows represent significant positive relationships and red arrows represent significant negative relationships. Arrows are depicted by and sized in proportion to the magnitude of their standardized regression coefficient. For simplification, only relationships with statistical significance based on full model analyses are depicted by arrows (74 plots within 40 sites). Source data are provided as a Source Data file.

## Abiotic influences on insect-mediated element fluxes

When evaluating the effects of abiotic variables on insect-mediated fluxes, MAT consistently contributed significantly and positively to H$_c$ and H$_i$ models (Figs. 2 and 3, Supplementary Table 4). In addition, soil C:N (CI = −0.35, −0.04) and soil C:P (CI = −0.47, −0.11) negatively correlated with H$_c$N and H$_c$P, respectively, while soil Si concentration (CI = 0.10, 0.48) correlated positively with H$_c$Si (Figs. 2 and 3). PET/MAP was not a significant contributor to variation in insect-mediated element fluxes at the global level (Fig. 2, Supplementary Table 4), but significantly explained variance of H$_c$N and H$_c$P between tropical forests and H$_c$N between temperate forests (Supplementary Table 5). Although annual contributions of elements (g m$^{-2}$ y$^{-1}$) derived from litter exceeded that of H$_c$ overall and especially in boreal forests, mean H$_c$P:Litter P in tropical forests was 1.8 times greater than that of temperate forests and 3.2 times greater than that of boreal forests (Supplementary Table 3).

Relative to other abiotic factors and in line with Hypothesis 2, MAT exerted the strongest positive influence on insect-mediated element fluxes (Fig. 2). Therefore, plant-insect herbivore interactions tend to interrupt efficient plant recycling of elements to a greater degree in warmer locations. We expect that insect-driven increases in the release of labile nutrients (insect frass, cadavers, greenfall, leachate, early abscised leaves) to the soil would exert strong effects on soil fertility and biogeochemical cycling, with the extent of these effects

dependent on site-specific characteristics[8,9]. We might also expect greater insect-mediated element fluxes to result from climate change-driven increases in insect outbreaks[33], which would amplify trends in insect-mediated fluxes from background herbivory. Soil C:N, soil C:P, and soil Si concentrations played additional roles for H$_c$, in support of Hypothesis 2, and also indicates that insect folivores move more nutrients in locations with nutrient-rich soils (Fig.2, Supplementary Table 5). In partial support of Hypothesis 2, PET/MAP and soil nutrient concentrations played significant roles for some insect-mediated nutrient fluxes within some latitude zones (Supplementary Table 5). Contrary to Hypothesis 2, however, PET/MAP did not have a significant effect on insect-mediated nutrient fluxes under non-outbreak conditions at the global level (Fig. 2). That the effects of PET/MAP and soil nutrient concentrations were inconsistent between scales and between latitude zones suggest that local to regional conditions can have an outsized influence over insect-mediated element fluxes (Fig. 2, Supplementary Table 5). For example, PET/MAP positively correlated with H$_c$N and H$_c$P in forests within the tropical zone (Supplementary Table 5), suggesting that overly wet conditions can reduce herbivory[34] and thus insect-mediated nutrient fluxes. Solar radiation and water availability can also be important drivers of insect activity during outbreaks, with droughts, which are typically local in scale, weakening tree defenses[35,36]. Finer scale measurements of temporal and spatial patterns such as seasonal and interannual precipitation or aspect may

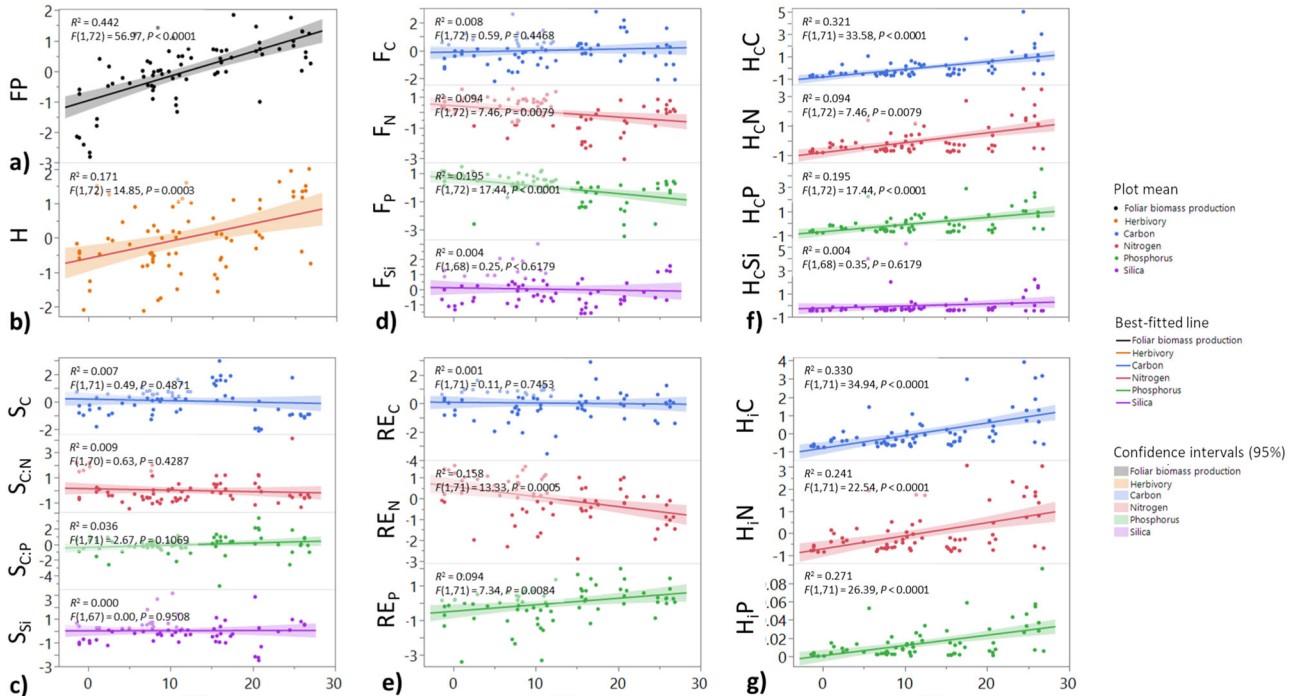

**Fig. 3 | Potential explanatory variables for insect-mediated element fluxes in broadleaved forests along a mean annual temperature (MAT) gradient. a** Foliar biomass production (FP, g m$^{-2}$ y$^{-1}$), (**b**) foliar herbivory rate (H, % leaf area removed y$^{-1}$), (**c**) soil nutrients or stoichiometry (S$_C$, S$_{C:N}$, S$_{C:P}$, S$_{Si}$; %), (**d**) foliar concentration of element E (F$_E$, %), (**e**) resorption efficiency of E (RE$_E$, %), (**f**) gross insect-mediated element flux (H$_c$, g m$^{-2}$ y$^{-1}$), and (**g**) net insect-mediated element flux

(H$_i$, g m$^{-2}$ y$^{-1}$) plotted against MAT. Response variables were log and logit transformed as necessary and (Z) standardized before regressing. Circles depict means within a forest plot (9–25 traps per plot), lines represent best-fitted lines, and bands are 95% CIs (74 plots). *F*-statistic, degrees of freedom, *P*-value, and *R*$^2$ value are reported for each simple regression (95% CI). Source data are provided as a Source Data file.

uncover differences in insect-mediated element fluxes generated by microclimatic variation[37,38]. High-resolution, locally sourced climate data might also better reveal potential roles of seasonal or more localized interannual water stress on insect-mediated element fluxes. Despite the coarse resolution of climate data for some sites, the results of our study demonstrate that MAT is a major driver of insect-mediated element fluxes, especially in nutrient-rich forests, with broad implications for large-scale biogeographic patterns in forest composition and function.

### Biotic influences on insect-mediated element fluxes (H$_c$ and H$_i$)

In estimating relative contributions of biotic variables to insect-mediated element fluxes, we found that foliar biomass production and herbivory explained more variation in H$_c$ and H$_i$ than did foliar element concentrations for C, N and P (Fig. 2, Supplementary Table 6). In contrast, foliar Si concentrations explained more variation in H$_c$Si than foliar biomass production or herbivory. Resorption efficiency played a smaller role than foliar biomass production or herbivory in explaining variation in H$_i$.

Because leaves in this study were passively collected in litter traps of a known area at the end of the leaf's life span, we were able to avoid some biases in estimating herbivory related to timing (e.g., snapshot versus repeated), scale (e.g., single versus multiple species), and selection (e.g., systematic versus haphazard) described in prior syntheses[6]. Our 4.12% global mean for foliar herbivory (Supplementary Table 2) is consistent with the 4.30% global estimate for herbivory of woody plant leaves in an extensive meta-analysis[16] when herbivory data were collected in a "blinded way" where leaf collectors were unaware of the hypothesis being tested. However, the variation of herbivory across our plots was substantial, with leaf area loss ranging from 0.6 ± 0.1% to 13.6 ± 0.8% (Supplementary Table 3).

Our results indicate a relatively consistent pattern of contribution by biotic precursors to insect-mediated element fluxes. The observed variability of insect-mediated nutrient fluxes appears to be explained primarily by foliar biomass production and herbivory, both of which were strongly linked to MAT (Fig. 2). In contrast to Hypothesis 3, however, foliar element concentrations played a smaller role in explaining H$_c$ variability across all plots relative to foliar biomass production and herbivory (Fig. 2). Variation in herbivory and foliar biomass production also played larger roles in shaping patterns of H$_i$ than did variability of foliar element concentration or resorption efficiency. For P, this might be explained in part as an influence of foliar element concentration being offset by the effects of resorption.

To the best of our knowledge, this is the largest study to investigate terrestrial Si cycling in natural systems, and our findings were somewhat equivocal. Although we detected a strong mass-balance driven relationship between foliar Si concentration and insect-mediated Si flux (Fig. 2d), we did not observe a significant relationship between foliar Si concentration and herbivory−indicating that foliar Si may play a minor role in plant defense against background levels of herbivory in mature, undisturbed broadleaf-dominated forests. Rather, the role that Si plays in plant herbivore defense may be overshadowed by biochemical defense strategies such as defensive investments into the production of alkaloid, terpene, or tannin compounds[39]. While plants may utilize Si for processes that lead to structural strength and for reducing other stressors such as drought[14,40], our results highlight that more work is needed to establish a central role for Si in plant herbivore defenses.

### Links between abiotic variables and biotic precursors to fluxes

Soil C:N (CI = −0.48, −0.18) was negatively correlated with foliar biomass production, whereas MAT (CI = 0.47, 0.83) was positively correlated with foliar biomass production (Figs. 2 and 3). Tropical forests

(Table 1) produced more foliar biomass than temperate forests, which produced more foliar biomass than boreal forests ($F_{(2, 71)} = 17.71$, $P < 0.001$). Only MAT (CI = 0.20, 0.67) positively correlated with herbivory globally (Figs. 2 and 3). Tropical forests (Table 1) exhibited greater herbivory than temperate or boreal forests ($F_{(2, 71)} = 5.22$, $P = 0.008$). While soil C:N (CI = −0.44, −0.00) and soil Si concentration (CI = 0.09, 0.62) were correlated with their respective foliar nutrient concentrations (Fig. 2), foliar P concentration positively correlated with MAT (CI = 0.10, 0.66; Fig. 3). Foliar N concentrations in tropical forests (Supplementary Table 3) were lower than those in temperate and boreal forests ($F_{(2, 71)} = 11.73$, $P = 0.003$). Foliar P concentrations (Supplementary Table 3) were also significantly lower in tropical than in temperate or boreal forests ($F_{(2, 71)} = 26.80$, $P < 0.001$). MAT (Figs. 2 and 3) positively correlated with N resorption efficiency (CI = 0.14, 0.62) while PET/MAP was negatively correlated with N resorption efficiency (CI = −0.48, −0.03). N resorption efficiency (Supplementary Table 3) was significantly greater in boreal forests than in tropical or temperate forests ($F_{(2, 70)} = 12.07$, $P = 0.002$) whereas P resorption was significantly greater in tropical forests than in temperate or boreal forests ($F_{(2, 70)} = 14.05$, $P < 0.001$). Soil C:P (Fig. 2) was positively correlated with P resorption efficiency (CI = 0.08, 0.44), which was significantly greater in tropical than in temperate or boreal forests ($F_{(2, 70)} = 6.79$, $P = 0.002$; Supplementary Table 3). Foliar Si concentration was positively and weakly correlated with foliar biomass production ($F_{(1, 67)} = 4.68$, $R^2 = 0.05$, $P = 0.034$) but was not significantly correlated to herbivory.

MAT and soil C:N explained 49% of the variability in foliar biomass production, where higher MAT and lower soil C:N were associated with higher foliar biomass production (Fig. 2). Additionally, only MAT helped to explain herbivory among variables we studied (Fig. 2). We might expect then that $H_c$ and $H_i$ would be greatest in the tropics, where foliar biomass production, herbivory and MAT are the greatest and soil C:N is the lowest (Table 1, Supplementary Fig. 2). The effects of abiotic factors on foliar element concentrations were mixed, where soil element concentration played a significant role in explaining variability in foliar element concentrations for all elements except C, and MAT positively correlated with foliar P concentration (Figs. 2 and 3). The results of this investigation are in line with a study on *Quercus garryana* which determined that climate and leaf traits explained variation in plant-herbivore interactions[37]. Ultimately, different plant life forms and species are likely to exhibit different strategies to cope with herbivory based on local conditions[37,38].

Interestingly, foliar nutrient content did not significantly correlate with herbivory in this study, indicating that nutrients measured in plants may not correspond to what is available to insect herbivores. For example, leaf age can affect leaf toughness and thus nutrient extraction efficiency[41]. Further, improving plant quality can benefit early stages of insect development but may not improve insect survivorship or escape from natural enemies[41]. Alternatively, other variables such as MAT (Figs. 2 and 3) or defense strategies not explored in this study (e.g., volatiles, leaf toughness) may have played greater roles than foliar nutrient concentrations in explaining variability in herbivory[39]. Interactions between nutrients and other variables could also explain the negative relationship between foliar nutrient concentrations and herbivory. For example, because P availability is generally lower in tropical biomes where soils are more weathered[21], low foliar P concentrations could simply be incidental to local nutrient availability in warmer climates where herbivory is greatest. Our study suggests that both broad-scale variables and local-scale conditions, as well as plant traits may all simultaneously shape plant-herbivore interactions. Future work would be needed to disentangle these effects.

P resorption efficiency was highest in tropical broadleaved forests while N resorption efficiency was highest in boreal forests, supporting the broadly appreciated and supported perspective that P limits tree growth in the tropics while N limits growth in higher latitudes (Supplementary Table 3)[22,42]. Furthermore, though soil C:N did not differ appreciably across biomes, soil C:P in tropical forests was significantly higher than in temperate and boreal forests, highlighting again the potential importance of labile compounds derived from insect herbivory in nutrient-limited systems (Supplementary Table 3). Our results are consistent with a recent meta-analysis on global-scale patterns of nutrient resorption of woody plants in which N resorption efficiency significantly decreased with MAT whereas P resorption efficiency marginally increased with MAT (Figs. 2 and 3)[43]. As with insect-mediated element fluxes, this suggests that although the extent of nutrient resorption efficiency might be regulated by local factors such as substrate[43], large-scale variables such as MAT can drive broad spatial patterns of ecosystem processes.

We investigated the effects of entire naturally occurring insect assemblages in mature, undisturbed broadleaved tree communities to develop an integrated description of insect herbivore-mediated element fluxes in tropical, temperate and boreal forests. We found compelling evidence that MAT plays a strong global role in insect-mediated element fluxes. The complexity embodied in this study, which includes an extraordinary diversity of plants representing a tremendous range of defense strategies interacting with diverse insect communities across sites with highly variable resource conditions, makes the generalization more remarkable. These results provide contemporary baseline data to better inform Earth system modeling on the myriad interactions among herbivores, plants, and soil. Next generation models could integrate these different processes and predict the overall effects of herbivores on ecosystem structure and function over longer time scales[44,45].

## Methods

### Study sites

We established a network of 74 measurement plots in mature, undisturbed forests at 40 sites across 6 continents (Supplementary Data 1). In the global network (Supplementary Fig. 1), 16 of the sites (33 plots) are tropical, 18 sites (32 plots) are temperate, and 6 sites (9 plots) are boreal[46], representing 12 of the 14 major broadleaved forest types[17].

The method used to assess stand-level herbivory and element flux relies on the visual estimation of leaf area removal from passively collected leaves[10]. Visual quantification of incremental removal from needle shaped leaves was not possible, so we focused our measurements on sites dominated by broadleaved tree species. That we examined only broadleaved-dominated forests also provided some level of constraint on the global study. To minimize the confounding influence of disturbance history, we set up plots in mature forest stands where there was no documented evidence or visible indications of recent human activity. All of the forests in this study could be classified as primary or old-growth forests[47]. Of the 74 forest plots, 68 plots spanned an area of 1 ha each, and 64 plots were situated at least 450 m from major anthropogenic disturbances such as major roads or settlements (Supplementary Data 1). During site selection, we also consulted local experts and literature for information on current and past natural (e.g., insect outbreaks) and anthropogenic (e.g., logging) histories (Supplementary Data 1). Plots ultimately included in the analysis did not show evidence of recent insect outbreak activity. Plots were not randomized, and some plots were in site clusters along elevation or precipitation gradients (Supplementary Data 1).

### Green leaf, leaf litter and soil collections

To characterize green leaf nutrients, we sampled and pooled 150 or more (to obtain at least 25 g dry weight) green, healthy (no to minimal signs of damage or discoloration) leaves at multiple heights in the canopy from at least ten randomly selected trees during the growing season for deciduous forest plots, and every dry and wet season for

evergreen forest plots. We dried all green leaves at 70 °C until constant mass and finely ground them for chemical analysis.

We installed 9–25 litter traps (0.1–0.5 m2 in area) in each of the 74 plots such that leaf litter traps were 0.3–1.0 m above the surface of the ground and spaced at ~20-m intervals (one site exception to interval distance is described in Supplementary Data 1). We collected litter for 1 or 2 years (between 2018 and 2021, see two site exceptions to year of collection described in Supplementary Data 1); for each plot, litter was collected every 14–31 days. In cases where 2 years of data were collected, we calculated an annual mean for regression analyses. We dried all leaves at 70 °C until constant mass and weighed them to estimate total leaf litterfall as g m$^{-2}$ y$^{-1}$. We then subsampled leaves from each litter trap and pooled ~160 leaves or more by plot (to obtain at least 25 g dry weight), which we then finely ground for chemical analysis.

To characterize soil, we homogenized ten cores (0–15 cm depth) once during the growing season at each plot. After sieving through 2 mm mesh, we dried soils at 65–70 °C until constant mass and ground them for chemical analysis. We report the soil nutrient concentrations for C and Si, and the soil stoichiometric ratios C:N and C:P derived from soil nutrient concentrations.

## Chemical analyses

For total C and N analyses, we combusted finely ground and homogenized green leaf, litter, and soil samples in an elemental analyzer at Copenhagen University, Denmark (Flash 2000, Thermo Scientific, Bremen, Germany). To estimate total P, we calculated the difference between the amounts of inorganic P extracted by sulfuric acid for ignited and unignited samples using an autoanalyzer (Seal AA500 Continuous Flow Analyzer; Seal Analytical GmbH, Norderstedt, DE) at Copenhagen University, Denmark[48]. To obtain Si concentrations, we digested samples in a sodium carbonate solution for 3 h (leaves) and 5 h (soils) before analyzing them with a SmartChem© 200 Discrete Analyzer (AMS Alliance, KPM Analytics, Westborough, MA, USA) at Lund University, Sweden[49]. In the case of the Malaysian and Brazilian sites, soil and foliar C, N, and P concentrations in this study were derived from previously published estimates (Supplementary Data 1).

## Estimates of foliar herbivory

We scanned and visually assessed insect damage to half or all collected broadleaved litter before processing for chemical analysis using a classification system described by Alliende[50]. A single observer scored insect damage according to six levels of leaf area removal by herbivores (0-1%, 1–5%, 5–25%, 25–50%, 50–75%, and >75%) from quarterly scans of leaf litter. This method for visually estimating damage and assessing leaf area consumption is efficient and accurate[6,51], with resulting estimates of herbivory from abscised leaves providing a metric of leaf-level insect defoliation accumulated over the entire lifespan of leaves. The approach provides an unbiased, community-level estimate of background level foliar herbivory.

## Calculations of foliar biomass production and insect herbivore-related element fluxes

To estimate gross and net insect-mediated release of elements from the canopy via foliar consumption, we used the approach described in Metcalfe et al.[10]. Specifically, we calculated this release as the product of: (i) foliar biomass production (FP), (ii) green leaf element concentration ($F_E$), (iii) foliar herbivory rate (H), and (iv) element resorption ($RE_E$, for net release estimates of C, N and P), where subscript E could be the elements C, N, P, or Si (Fig. 1).

We estimated annual leaf production at each plot using 1 or 2 years of litterfall collections and calculated nutrient fluxes due to insect defoliation from these estimates. We converted the total dry mass of leaf litter accumulated over the year divided by known trap area ($L_H$, g m$^{-2}$ y$^{-1}$) to annual foliar biomass production (FP) calculated as $L_H/(1-H)$, where H is the proportion of leaf litter removed by insect folivores. We converted FP to element production with foliar element concentration ($F_E$) data. $F_E$ was then multiplied by H, yielding total leaf elements consumed by insect herbivores ($H_c$), or the gross insect-mediated element flux (see Supplementary Table 1 for equations, based on methods described in Metcalfe et al.[10]). We assume $H_c$ is approximately equal to the quantities of the same elements released by herbivores to the ground via excreta, bodies, moults, and unconsumed leaf fragments under steady-state assumptions, which should be reasonable for these relatively intact mature forests where herbivore populations, and the balance between ingoing and outgoing migratory herbivores, should be relatively stable over time.

To further understand the potential importance of the herbivory-mediated fluxes resulting from insects intercepting green leaves before resorption (Fig. 1), we also estimated resorption efficiency (% withdrawal of an element during senescence) of C, N and P following Vergutz et al.[52]. That is, we utilized mass loss correction factors 0.78 and 0.784 for predominantly evergreen angiosperm forests and predominantly deciduous angiosperm forests, respectively[52]. Using this approach, we assumed nominal nutrient leaching from litterfall traps between collections and that herbivores exclusively target foliage before initiation of resorption. By combining $H_c$ with plot-level resorption estimates for C, N and P, we estimated this net insect-mediated element flux ($H_i$)— defined here as the additional C, N or P released via green leaf herbivory from plants to soil prior to senescence-related resorption (Fig. 1; Supplementary Table 1). That is, $H_i$ is the difference in total (litter + herbivore-mediated) nutrient inputs between a scenario with herbivores and a scenario without herbivores (Fig. 1). $H_c$ is the total amount of a foliar element consumed and ultimately released to the ecosystem by herbivores. If we assume that leaves abscise at the end of their lifetime after resorption has occurred, and that the vast majority of herbivory occurs before this resorption[26], then $H_i$ is designed to calculate the foliar elements released by herbivores that would not otherwise have been released via litterfall. For this reason, $H_i$ is more closely comparable to external inputs of relatively labile, inorganic nutrients derived from atmospheric deposition, biological fixation or bedrock weathering. In this analysis, we do not distinguish between C allocated to biomass versus respiration by herbivores because this allocation is poorly constrained for most geographies, biome types, and herbivore groups[10,53]. We report only gross ($H_cC$) and net ($H_iC$) C removed from the foliage by herbivory.

## Estimating other major sources of labile nutrients

To compare the amounts of elements passing through insect folivores with other major sources of labile nutrients, we derived atmospheric N and P-values from models developed by Brahney et al.[54] and we estimated mineral-weathered P-values from the model described by Hartmann et al.[55].

## Climate variables

To obtain air temperature estimates, we installed TMS-4 dataloggers (Tomst S.R.O, Czech Republic) at the plots during the collection period. In case of gaps in temperature data, we supplemented our dataset with other local measurements (e.g. on-site instruments from other researchers, local weather station data), scientific literature, and consulted with local experts to finalize MAT estimates (Supplementary Data 1). Dryness or climate ratios[56,57] depict the ratio of potential evapotranspiration (PET) to mean annual precipitation (MAP). We derived mean annual PET at each plot after summing the monthly means (Penman-Monteith method) for each year during the 2018–2021 collection period using the dataset from Singer et al.[57]. In the few cases where collections were made during different times (2017 for the Malaysian site, 2010 for the Brazilian site), we obtained PET for their associated years[57]. We consulted local weather station data, scientific literature, and local experts to estimate MAP for all plots (Supplementary Data 1).

**Table 3 | Main variables used in pathway diagram**

| Category | Variables |
|---|---|
| Abiotic factors | $S_C$, $S_{C:N}$, $S_{C:P}$, $S_{Si}$, MAT, PET/MAP |
| Biotic factors | FP, H, $F_E$, $RE_E$ |
| Insect-mediated element fluxes | $H_GE$, $H_iE$ |

Construction grouped by category: soil C concentration ($S_c$), soil C:N ($S_{C:N}$), soil C:P ($S_{C:P}$), soil Si ($S_{Si}$) concentration, mean annual temperature (MAT), dryness as potential evapotranspiration: mean annual precipitation (PET/MAP), foliar production (FP), foliar herbivory (H), foliar concentration of element E ($F_E$), resorption efficiency of E ($RE_E$), gross insect-mediated element flux of E ($H_GE$), and net insect-mediated element flux of E ($H_iE$). Each category row is assumed to have a potential causal effect on lower rows.

## Statistics and reproducibility

We applied linear mixed models to assess global patterns in insect herbivore-mediated fluxes across latitude using R programming language v. 4.2.1 computer[58]. To rule out severe multicollinearity, we ensured that variance inflation factors (vif function in the car v. 3.1–1 package) for all predictors fell below 2 before proceeding with each model[59]. We fitted the linear mixed models in the lme4 v. 1.1–030 package[60], where site or site:plot was treated as a random factor for all models to account for unique geographical and floristic characteristics. Because we did not detect a significant difference in herbivory across latitude between northern and southern hemispheres in this dataset, we converted latitude to absolute values. We first determined the effect of abiotic variables on biotic variables with simple models (single fixed variable + site as random factor). The main variables used to construct full models (all fixed variables + site as random factor) are summarized in Table 3. Before fitting the full models with correlated variables from simple regressions, we produced standardized regression coefficient values for the global analysis by converting all variables to z scores across each dataset after log- or logit-transforming variables as necessary. However, we could not fully resolve moderate unequal variances in the full model for foliar biomass production[61]. We constrained 95% confidence intervals (CI) of the effect sizes with 1000 parametric bootstrap simulations to obtain a conservative estimate of significance[62] and considered effects significant when CI did not cross zero. We report marginal and conditional $R^2$ GLMM values as goodness-of-fit statistics using the MuMIn v. 1.47.1 package[63]. We constructed pathway diagrams between abiotic, biotic precursor, and insect-mediated element fluxes to summarize results of the linear mixed models and depict potential causal relationships.

To test for differences in variable means between latitude zones, we performed two-way Kruskal-Wallis non-parametric tests (kruskal.test function in stats v. 4.2.1 package) followed by Dunn pairwise comparisons with Bonferroni-Holm adjustments (dunnTest function in FSA v. 0.9.5 package).

Though observations in this study were likely dependent on spatial and temporal context and were thus difficult to reproduce, computational reproducibility can be achieved by using the same datasets, codes and software as this study. As a result of shared workflow, transparency allows for other researchers to build directly on this primary work.

## Reporting summary

Further information on research design is available in the Nature Portfolio Reporting Summary linked to this article.

## Data availability

The data generated in this study have been deposited to the online repository figshare[64]. The source data are provided in the Source Data file. Additional source data are referenced in the Supplementary Data 1 file. Source data are provided with this paper.

## Code availability

R code generated for the current study has been deposited to the online repository, figshare[64].

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

## Acknowledgements

This project was primarily funded by the European Research Council under the European Union's Horizon 2020 research and innovation pro-gramme (ECOHERB; grant no. 682707—D. M.). Additional funding for work in Chile was supported by Proyecto Fondecyt 1130898—M. J. Funding in Kevo, Finland and Khanty-Mansysk, Russia was partially

funded by the European Research Council under the European Union's Horizon 2020 research and innovation programme (EU-INTERACT III; grant no. 871120—B. H.). Plots in the central Congo were funded by a Natural Environment Research Council Large Grant (CongoPeat; grant no. NE/R016860/1—S. L.). The plot in Udzungwa, Tanzania was funded by an Australian Research Council Future Fellowship (grant no. FT170100279—A. M.) and the University of the Sunshine Coast. The USDA Forest Service, Pacific Southwest Research Station provided additional financial and logistical support for work in Hawai'i. The participation of P. Š. was partly covered by institutional, non-project support from VUKOZ. We sincerely thank Jens Hartmann from the University of Hamburg, Germany and Natalie Mahowald from Cornell University in New York, USA. Many thanks to Per Lennart Ambus and Søs Ludvigsen at the University of Copenhagen, Denmark as well as Daniel Conley and Isa Doverbratt at Lund University, Sweden. We thank Creighton Litton from the University of Hawai'i at Mānoa as well as Peter Vitousek from Stanford University in California, USA. Thanks to Kaianui Andaya, Alealani Evangelista, Chantilly Keliihoomalu, Chloe Martins-Keliihoomalu, Rebekah Ohara, Cody Pacheco, Kirie Santos, Naneaikealaula Thomas, Kauilamakahikina Thomas, and Tyler Uehara with the USDA Institute of Pacific Islands Forestry, the Akaka Foundation for Tropical Forests, and the Kupu Conservation Leadership Development Program in Hawai'i, USA. Thanks to Michelle Schiffer and the support staff at the Daintree Rainforest Observatory as well as Lucas Cernusak, Susan Laurance, and Michael Liddell at James Cook University Cairns in Australia. Thanks also to Lindy Orwin, Kevin Jackson, and the Cooloola CoastCare team in Queensland, Australia. We also thank Guanhua Dai, Xiuxiu Wang, and Hao Xu at the Research Station of Changbai Mountain Forest Ecosystems, Chinese Academy of Sciences, as well as Jiabing Wu at Institute of Applied Ecology, Chinese Academy of Sciences. Thanks to Huangfenglin at the Jianfengling site in Hainan, China. We thank Erick Materu from the KiLi Group in Moshi, Tanzania, as well as Kelvin Kumasi, Revocatus Raulien, and the crew from the Forest Research Institute in Ghana. Thanks also to Otso Suominen from the Kevo Research Station in Finland and Vasile Sebastian Stătescu from Transylvania University in Romania. Thanks to Kamil Král from Silva Tarouca Research Institute in Czech Republic and Johan Eckdahl from Lund University in Sweden. We thank Terhi Riutta in Borneo. Many thanks to James Costa, Carson Ellis, Thomas Hennesy, Jason Love, Angela Faye Martin, and Brent Martin at the Highlands Biological Station and Western Carolina University in North Carolina, USA. Thanks also to Samuel Rose in Colorado, USA, Stéphane Bourassa and Benoît Lafleur in Quebec, Canada, as well as Matthew Stewart and Neville Winchester in Vancouver, Canada. We thank Vladimir Aleksandrovich Oskolkov in Irkutsk, Russia and the Baikal State Natural Biosphere Reserve. We also thank Maria Edith in Cusco, Peru. Thanks to Trésor Ahougoni, Yannick Bocko Michel Iwango, Felix Koubouana, Joel Loumeto, Richard Molayi, René Mpoukou, and Guy Ngbongo from Marien Ngouabi University in the Congo. We are grateful to the government of the Republic of the Congo and Wildlife Conservation Society Congo for logistical support.

## Author contributions

B. C. Hwang and D. B. Metcalfe designed the study. S. Abu-Bredu, T. Andersson, J. Calvo-Alvarado, M. Churski, M. Jiménez-Castillo, G. C. Dargie, H. Diao, V. G. Duboscq-Carra, N. Filippova, S. A. Gage, M. Gaglianese-Woody, D. Galiano-Cabrera, A. Hemp, W. H. Huasco, B. C. Hwang, G. Iankoshvili, A. V. Ivanov, M. A. Kaswamila, D. Kuijper, P. Lobos-Catalán., H. Lyatuu, Y. E. Mampouya Wenina, B. Materu, M. Mbemba, R. Moritz, L. Mumladze, K. Orang, A. C. Palma, I. C. Petritan, S. Plyusnin, B. L. Puma Vilca, M. Rodríguez-Solís, M. Rodriguez-Cabral, P. Šamonil., K. Stępniak, I. A. Suspense, and S. K. Walsh performed work in the field and lab. B. C. Hwang and D. B. Metcalfe managed the work with logistical support from J. Calvo-Alvarado, M. N. Barrios-Garcia, M. Jiménez-Castillo, K. S. Francisco, C. P. Giardina, C. Hemp, D. Kuijper, S. L. Lewis, Y. Malhi, A. R. Marshall, A. S. K. Ngute, I. A. Suspense, H. Xu, and A. Zagidullina. B. C. Hwang analyzed the data with support from N. G. Johnson; B. C. Hwang led the writing of the manuscript. C. P. Giardina and D. B. Metcalfe led the revisions with input from S. Abu-Bredu, J. Calvo-Alvarado, M. N. Barrios-Garcia, G. C. Dargie, H. Diao, V. G. Duboscq-Carra, A. Hemp, C. Hemp, W. H. Huasco, A. V. Ivanov, N. G. Johnson, D. Kuijper, P. Lobos-Catalán, S. L Lewis., Y. Malhi, A. R. Marshall, L. Mumladze, A. S. K. Ngute, A. C. Palma, I. C. Petritan, M. Rodriguez-Cabral., I. A. Suspense, and A. Zagidullina.

## Funding

## Competing interests

The authors declare no competing interests.

## Additional information

**Bernice C. Hwang** [1,2,3 ✉], **Christian P. Giardina** [4], **Stephen Adu-Bredu** [5,6], **M. Noelia Barrios-Garcia** [7,8], **Julio C. Calvo-Alvarado** [9], **Greta C. Dargie** [10], **Haoyu Diao** [11,12], **Virginia G. Duboscq-Carra** [13], **Andreas Hemp** [14], **Claudia Hemp** [14,15], **Walter Huaraca Huasco** [16,17], **Aleksandr V. Ivanov** [18], **Nels G. Johnson** [19], **Dries P. J. Kuijper** [20], **Simon L. Lewis** [10,21],

Paulina Lobos-Catalán [22], Yadvinder Malhi [16], Andrew R. Marshall[23,24], Levan Mumladze [25], Alain Senghor K. Ngute [23], Ana C. Palma [26], Ion Catalin Petritan [27], Mariano A. Rordriguez-Cabal[7,13], Ifo A. Suspense[28,29], Asiia Zagidullina[30,31], Tommi Andersson [32], Darcy F. Galiano-Cabrera[17,33], Mylthon Jiménez-Castillo[22], Marcin Churski [20], Shelley A. Gage[34], Nina Filippova[35], Kainana S. Francisco [4], Morgan Gaglianese-Woody[36], Giorgi Iankoshvili [37], Mgeta Adidas Kaswamila[15], Herman Lyatuu[24], Y. E. Mampouya Wenina[28,29], Brayan Materu[15], M. Mbemba[38], Ruslan Moritz [39], Karma Orang[40], Sergey Plyusnin[41], Beisit L. Puma Vilca [17,32], Maria Rodríguez-Solís [9], Pavel Šamonil[42], Kinga M. Stępniak [20,43], Seana K. Walsh [44], Han Xu[45] & Daniel B. Metcalfe[1,2]

[1]Department of Physical Geography and Ecosystem Science, Lund University, Lund, Sweden. [2]Department of Ecology and Environmental Science, Umeå University, Linnaeus väg 6, Umeå, Sweden. [3]Department of Ecology, University of Innsbruck, Sterwartestraße 15, Innsbruck, Austria. [4]Institute of Pacific Islands Forestry, Pacific Southwest Research Station, USDA Forest Service, Hilo, HI, USA. [5]CSIR-Forestry Research Institute of Ghana: Kumasi, Ashanti, Ghana. [6]Department of Natural Resources Management, CSIR College of Science and Technology, Kumasi, Ghana. [7]Rubenstein School of Environment and Natural Resources, University of Vermont, Burlington, VT 05405, USA. [8]CONICET, CENAC-APN, Universidad Nacional del Comahue (CRUB), Bariloche (8400), Argentina. [9]Escuela de Ingeniería Forestal, Tecnológico de Costa Rica, Cartago, Costa Rica. [10]School of Geography, University of Leeds, Leeds, UK. [11]CAS Key Laboratory of Forest Ecology and Management, Institute of Applied Ecology, Chinese Academy of Sciences, Shenyang 110016, China. [12]Swiss Federal Institute for Forest, Snow and Landscape Research WSL, Birmensdorf 8903, Switzerland. [13]Grupo de Ecología de Invasiones, Instituto de Investigaciones en Biodiversidad y Medioambiente (INIBIOMA)—CONICET—Universidad Nacional del Comahue, Bariloche, Argentina. [14]Department of Plant Systematics, University of Bayreuth, Bayreuth, Germany. [15]Senckenberg Biodiversity and Climate Research Centre, Frankfurt, Germany. [16]Environmental Change Institute, School of Geography and the Environment, University of Oxford, Oxford OX1 3QY, UK. [17]Asociación Civil Sin Fines De Lucro Para La Biodiversidad, Investigación Y Desarrollo Ambiental En Ecosistemas Tropicales (ABIDA), Urbanización Ucchullo Grande, Avenida Argentina F-9, Cusco, Perú. [18]Institute of Geology and Nature Management Far Eastern Branch of Russian Academy of Sciences, Relochny lane, 1, Blagoveshchensk 675000, Russia. [19]Pacific Southwest Research Station, USDA Forest Service, Hilo, Hawai'i, USA. [20]Mammal Research Institute, Polish Academy of Sciences, Ul. Stoczek 1, 17-230 Białowieża, Poland. [21]Department of Geography, University College London, London, UK. [22]Instituto de Ciencias Ambientales y Evolutivas, Universidad Austral de Chile, Campus Isla Teja, Valdivia, Chile. [23]Forest Research Institute, University of the Sunshine Coast, Sippy Downs, Queensland, Australia. [24]Reforest Africa, PO Box 5 Mang'ula, Kilombero District, Tanzania. [25]Institute of Zoology, Ilia State University, 3/5 Cholokashvili Ave, 0169 Tbilisi, Georgia. [26]College of Science & Engineering and Centre for Tropical Environmental and Sustainability Science, James Cook University, Qld, Australia. [27]Faculty of Silviculture and Forest Engineering, Transilvania University of Brașov, Șirul Beethoven 1, 500123 Brașov, Romania. [28]Ecole Nationale Supérieure d'Agronomie et de Foresterie, Université Marien Ngouabi, Brazzaville, République du Congo. [29]Laboratoire de Biodiversité, de Gestion des Ecosystèmes et de l'Environnement, Faculté des Sciences et techniques, Université Marien Ngouabi, Brazzaville, République du Congo. [30]Forest Research Institute, University of Quebec in Abitibi-Témiscamingue, QC, Canada. [31]Department of Physical Geography and Environmental Management Problems, Institute of Geography, Russian Science Academy, Moscow, Russia. [32]Kevo Subarctic Research Institute, Biodiversity Unit, University of Turku, 20014 Turku, Finland. [33]Facultad de Ciencias Biológicas, Universidad Nacional de San Antonio Abad del Cusco, Av. de La Cultura 773, Cusco, Cusco Province 08000, Peru. [34]Centre for Horticultural Science, Queensland Alliance for Agriculture and Food Innovation, The University of Queensland, 47 Mayers Road, Nambour 4056, Australia. [35]Yugra State University, 628012Chekhova street, 16, Khanty-Mansiysk, Russia. [36]Appalchian State University, 572 Rivers Street, Boone, NC 28608, USA. [37]Institute of Ecology, Ilia State University, 3/5 Cholokashvili Ave, 0169 Tbilisi, Georgia. [38]CongoPeat Project, Ecole Nationale Supérieure d'Agronomie et de Foresterie, Université Marien Ngouabi, Brazzaville, République du Congo. [39]Siberian Institute of Plant Physiology and Biochemistry SB RAS, 664033 IrkutskLermontova str., 132Russia. [40]Ugyen Wangchuk Institute for Forest Research and Training, Department of Forests and Park Services, Ministry of Energy and Natural Resources, Lamai Goempa, Bumthang, Bhutan. [41]Pitirim Sorokin Syktyvkar State University, 455 Oktyabrsky prosp., 167001 Syktyvkar, Russia. [42]The Silva Tarouca Research Institute, Květnové náměstí 391, Průhonice 252 43, Czech Republic. [43]Department of Ecology, Faculty of Biology, University of Warsaw, Żwirki i Wigury 101, 02-086 Warsaw, Poland. [44]Department of Science and Conservation, National Tropical Botanical Garden, 3530 Papalina Road, Kalāheo, HI 96741, USA. [45]Research Institute of Tropical Forestry, Chinese Academy of Forestry, Guangzhou 510520, China. ✉e-mail: bernice.hwang@uibk.ac.at

