## [Peer Review File · Nature Communications]

The impact of insect herbivory on biogeochemical cycling in broadleaved forests varies with temperatureREVIEWER COMMENTS

Reviewer #1 (Remarks to the Author):

This study addressed the variations of nutrient fluxes associated with background-level insect herbivory in global broadleaved forest ecosystems with standardized methods. Authors reported that mean annual temperature is an important abiotic factor in driving the insect herbivory mediated nutrient fluxes across tropical, temperate, and boreal forests. While I found the results are meaningful, I have several concerns on this work.

The first hypothesis (H1) is not appropriate. There are many nutrient fluxes in forest ecosystems. While insect-mediated nutrient flux is important, it cannot exceed all other major fluxes. For example, the nutrient absorption of plants from soil will be much higher than any other fluxes. At least, it is not clear for the statement 'other major fluxes' in H1.

For the second hypothesis (H2), authors did not provide enough background information. For example, the core words 'temperature', 'dryness', and 'soil nutrient concentration' actually did not present in the earlier paragraphs of the Introduction section.

For the third hypothesis (H3), it is clearly that this is a hypothesis based on the results.

Line 1 As this study collected samples from broadleaved forests, it is important to show the information of 'broadleaved' in the title.

Line 96 Many ecosystem processes would be altered by herbivorous. It would be better to clearly state which process this study will focus on.

Line 104 It is not clear for the meaning of 'Insect-mediated nitrogen fluxes'

Line 109-111 The findings from large spatial scale, such as this study, would have limited implications for the temporal changes of temperature at local scale. The same process (as insect herbivory in this study) would be controlled by quite different ecological factors at

global and local scales, and between temporal and spatial scales.

Line 143 What do you mean by 'deposition'?

Line 228 Table 2. Please check the value and the unit of mineral weathered P. Are more than 50 g P m⁻² being mineralized annually for all the forests averaged globally?

Figure 2. P values for the regressions should be shown in the figure.

Line 289-290 and Figure 2d, If foliar N was higher in tropical forests than other two forest types (Line 289-290), why F_n was negatively correlated with MAT as shown in Figure 2d?

The similar problem with F_p (figure 2d and Lines 290-291).

Line 306-307 The role of insect-mediated fluxes is overstated, as authors only compared that with one flux (atmospheric nutrient deposition) in Table 1. Actually, only in 20% location the insect-mediated N fluxes was higher than atmospheric N deposition, that means insect mediated N flux in most sites was much lower than N deposition.

Line 315-316 Reference is needed for this statement.

Line 352-353 the impacts of in situ warming would be hardly inferred from space-time substitution.

Line 480 broadleaved forests

Reviewer #2 (Remarks to the Author):

This is a very interesting study where leaf samples from forests all over the world were collected and analyzed for chemical composition and herbivory. Soil chemical analysis was also performed and the impact of "background herbivory" on element cycling was calculated. The amount of data is impressive and I consider that the message that "background herbivory" has a significant impact on element cycling is important and interesting for a broad audience.

I am not an expert on insect herbivory effects on element cycling in forests but I consider this work original. Overall the work supports the conclusions but I have a few points that can be addressed in a possible revision.

First, the confidence intervals for figure 3, (named figure 3 erroneously) are extremely small. This seems wrong as they are supposed to be 95% C.I. according to the legend. also, the R²

is very low in a number of graphs and i wonder whether these relationships are significant. it is not presented.

Second, I have a question about the calculations. both the Hc and Hi are based on calculations, and not measured in real life. What conclusions can be drawn from the Hi values, if these are calculations, and do you also have information from some sites based on actual nutrient cycling data? Herbivory now in the SEM leads to Hc and Hi, but because this is based on calculations, with the same data, in particular the Hi link is questionable I think.

Third, leaf N is measured after drying at 70%, but it is well described that the N concentration is reduced when drying at such high temperature, and that N is usuallyt measured after drying at lower temperatures, max 60 C i believe.

Line 185 says: we anticipate that soil nutrient concentration (SE) remains unchanged with H, even as element cycling increases. I wonder if this is true. If i understand it correctly, nutrient cycling is thought to be affected but it does not change the actual concentration in the soil, ie all the nutrients are taken up? Why is this true and why is this important. Please explain.

Minor comment: values in Appendix D are not presented in percentages (e.g. for leaf N) even though it says so in the table

RESPONSE TO REVIEWER COMMENTS

Reviewer #1 (Remarks to the Author):

(Comment) This study addressed the variations of nutrient fluxes associated with background-level insect herbivory in global broadleaved forest ecosystems with standardized methods. Authors reported that mean annual temperature is an important abiotic factor in driving the insect herbivory mediated nutrient fluxes across tropical, temperate, and boreal forests. While I found the results are meaningful, I have several concerns on this work.

(Response) We thank the editor and reviewers for their generally positive appraisal of the importance and broad interest of the manuscript topic, and their comments on the text. We have now revised the manuscript in response to these comments, which has further improved the manuscript. We believe that the resulting manuscript provides important insights on herbivores that will guide further research and help refine biogeochemical modelling efforts.

(Comment) The first hypothesis (H1) is not appropriate. There are many nutrient fluxes in forest ecosystems. While insect-mediated nutrient flux is important, it cannot exceed all other major fluxes. For example, the nutrient absorption of plants from soil will be much higher than any other fluxes. At least, it is not clear for the statement ‘other major fluxes’ in H1.

(Response) Indeed, there are numerous internal ecosystem fluxes of nutrients. In contrast to litterfall, many other fluxes are derived from relatively recalcitrant organic forms (e.g., soil organic matter) so flux rates from these pools are low. As a result, they have less potential to affect plant/ecosystem processes, relative to insect derived inputs, which like the typically larger litterfall flux, tend to be in more labile forms (Lovett et al., 2002). To clarify, we have revised H1 to specify the fluxes to which we compare studied insect-mediated fluxes: “H1 – The flux of N and P from plant to soil mediated by insect herbivores would meet or exceed other major labile fluxes of these elements, such as atmospheric deposition (for N and P) and bedrock weathering (for P). (Lines 162-164) We have also mentioned these specific fluxes earlier in the introduction: “Furthermore, relative to leaf litter, herbivory-related insect deposits are typically enriched with labile forms of nutrients, and in many forests, insect-mediated nutrient fluxes are comparable to or even exceed

fluxes from other inputs of relatively labile, mineral forms of nutrients such as atmospheric deposition or bedrock weathering^{9,10}.” (Lines 134-137)

Lovett, G.M., Christenson, L.M., Groffman, P.M., Jones, C.G., Hart, J.E. and Mitchell, M.J., 2002. Insect defoliation and nitrogen cycling in forests: laboratory, plot, and watershed studies indicate that most of the nitrogen released from forest foliage as a result of defoliation by insects is redistributed within the ecosystem, whereas only a small fraction of nitrogen is lost by leaching. *BioScience*, 52(4), pp.335-341.

(Comment) For the second hypothesis (H2), authors did not provide enough background information. For example, the core words ‘temperature’, ‘dryness’, and ‘soil nutrient concentration’ actually did not present in the earlier paragraphs of the Introduction section.

(Response) We have now introduced the terms “mean annual temperature”, “dryness”, and “soil nutrient concentration” to the existing background information earlier in the introduction. The section now reads: “High nutrient availability, warm temperatures, and low water stress can positively influence foliar nutrient concentrations, foliar biomass production, and insect abundance with positive synergistic effects on overall insect-mediated element fluxes^{17,18,19}. Consequently, warm mean annual temperatures and high soil nutrient concentrations could increase insect herbivory and thus insect-mediated element fluxes in nutrient-limited but not water-limited (i.e., low dryness) systems.” (Lines 155-160)

(Comment) For the third hypothesis (H3), it is clearly that this is a hypothesis based on the results.

(Response) We have now revised this hypothesis to clarify our reasoning, and how it is independent of the findings in H1 and H2. New text in the introduction describes how the precursors to H_c/H_i tend to respond in the same direction to warming/moisture/fertility (Lines 155-160). H3 aims to assess how each of the precursors to H_i/H_c respond to these factors, which help to explain the broader results of H2. The new text outlining H3 is as follows: “Observed responses of insect mediated element fluxes to MAT, dryness and soil nutrient concentrations would be driven in equal measure by similar responses from foliar herbivory, biomass production and foliar element concentrations.” (Lines 168-170)

(Comment) Line 1 As this study collected samples from broadleaved forests, it is important to show the information of ‘broadleaved’ in the title.

(Response) We thank the reviewer for the title suggestion. We have altered the title to read: “Temperature mediates insect herbivore impact on biogeochemical cycling in broadleaved forests”. (Lines 1-2)

(Comment) Line 96 Many ecosystem processes would be altered by herbivorous. It would be better to clearly state which process this study will focus on.

(Response) We have replaced “ecosystem processes” with “biogeochemical cycling”. (Line 97)

(Comment) Line 104 It is not clear for the meaning of ‘Insect-mediated nitrogen fluxes’

(Response) We have replaced “Insect-mediated nitrogen fluxes” with “Nitrogen that was released from plants by insect herbivores”. (Lines 104-105)

(Comment) Line109-111 The findings from large spatial scale, such as this study, would have limited implications for the temporal changes of temperature at local scale. The same process (as insect herbivory in this study) would be controlled by quite different ecological factors at global and local scales, and between temporal and spatial scales.

(Response) There is a long and well-established history of using latitudinal variation to make inferences about ecosystem responses to temperature (e.g., De Frenne et al., 2013). However, we acknowledge that the primary strength and focus of our data lies in documenting and explaining controls over global biogeographical patterns in ecosystem processes, rather than inferring temporal change. We have rephrased the abstract and other relevant sections of the text to make this clearer (Lines 325-327, 354-355). The relevant section in the abstract has been revised to the following: “Further, these results show how climate affects interactions between natural populations with important consequences for biogeochemical cycles.” (Lines 110-112)

De Frenne P, Graae BJ, Rodríguez-Sánchez F, Kolb A, Chabrierie O, Decocq G, De Kort H, De Schrijver A, Diekmann M, Eriksson O, Gruwez R. Latitudinal gradients as natural laboratories to infer species' responses to temperature. *Journal of Ecology*. 2013 May; 101(3):784-95.

(Comment) Line 143 What do you mean by ‘deposition’?

(Response) To avoid confusion, we have chosen to replace “deposition” with “insect deposits” from the sentence which now reads: “Though global analyses on herbivory exist^{e.g.,16}, the flux of

nutrients associated with insect herbivory and insect deposits is poorly understood⁸, as is the potential impact of climate on these fluxes.” (Lines 143-145)

(Comment) Line 228 Table 2. Please check the value and the unit of mineral weathered P. Are more than 50 g P m⁻² being mineralized annually for all the forests averaged globally? Figure 2. [sic] P values for the regressions should be shown in the figure.

(Response) We thank the reviewer for pointing out the error; we have made the units correction in Table 2. (Line 237) We have also added p-values and other regression statistics to Fig. 3. (Line 314)

(Comment) Line 289-290 and Figure 2d, If foliar N was higher in tropical forests than other two forest types (Line 289-290), why F_n was negatively correlated with MAT as shown in Figure 2d? The similar problem with F_p (figure 2d and Lines 290-291).

(Response) We thank the reviewer for pointing out these discrepancies. We have corrected the text to read: “Foliar N concentrations in tropical forests (Appendix D) were lower than those in temperate and boreal forests ($F_{(2, 71)} = 11.73, P = 0.003$). Foliar P concentrations (Appendix D) were also significantly lower in tropical than in temperate or boreal forests ($F_{(2, 71)} = 26.80, P < 0.001$).” (Lines 401-404)

(Comment) Line 306-307 The role of insect-mediated fluxes is overstated, as authors only compared that with one flux (atmospheric nutrient deposition) in Table 1. Actually, only in 20% location the insect-mediated N fluxes was higher than atmospheric N deposition, that means insect mediated N flux in most sites was much lower than N deposition.

(Response) Indeed, we have claimed only partial agreement with our hypothesis. However, we have rephrased the sentence to further avoid overstating our results: “In partial agreement with Hypothesis 1, net insect-mediated nutrient fluxes from this pathway exceeded that of atmospheric nutrient deposition in only some localities (Table 1).” (Lines 241-243)

(Comment) Line 315-316 Reference is needed for this statement.

(Response) We have added reference 22 to the statement. (Line 252)

(Comment) Line 352-353 the impacts of in situ warming would be hardly inferred from space-time substitution.

(Response) This section has been revised in response to your earlier, related comment. While there is a long and well-established history of using latitudinal variation to make inferences about ecosystem responses to temperature (De Frenne et al., 2013), we acknowledge that the primary strength and focus of our data lies in documenting and explaining controls over global biogeographical patterns in ecosystem processes, rather than inferring temporal change. We have now revised relevant text in this section to make this clearer, as follows: “Therefore, plant-insect herbivore interactions tend to interrupt, otherwise highly efficient, plant recycling of elements to a greater degree in warmer locations.” (Lines 325-327)

De Frenne P, Graae BJ, Rodríguez-Sánchez F, Kolb A, Chabrerie O, Decocq G, De Kort H, De Schrijver A, Diekmann M, Eriksson O, Gruwez R. Latitudinal gradients as natural laboratories to infer species' responses to temperature. *Journal of Ecology*. 2013 May; 101(3):784-95.

(Comment) Line 480 broadleaved forests

(Response) We thank the reviewer for pointing out this typo. The sentence now reads: “Visual quantification of incremental removal from needle shaped leaves was not possible, so we focused our measurements on sites dominated by broadleaved tree species.” (Lines 474-476)

Reviewer #2 (Remarks to the Author):

(Comment) This is a very interesting study where leaf samples from forests all over the world were collected and analyzed for chemical composition and herbivory. Soil chemical analysis was also performed and the impact of "background herbivory" on element cycling was calculated. The amount of data is impressive and I consider that the message that "background herbivory" has a significant impact on element cycling is important and interesting for a broad audience.

I am not an expert on insect herbivory effects on element cycling in forests but i consider this work original. Overall the work supports the conclusions but I have a few points that can be addressed in a possible revision.

(Response) We thank the reviewer for a generally positive appraisal of the breadth of work and interest as well as the importance of the manuscript topic. Our responses to specific comments follow.

(Comment) First, the confidence intervals for figure 3, (named figure 3 erroneously) are extremely small. This seems wrong as they are supposed to be 95% C.I. according to the legend. also, the R^2 is very low in a number of graphs and i wonder whether these relationships are significant. it is not presented.

(Response) We thank the reviewer for pointing out the discrepancy. We have changed the figure name from Fig. 2 to Fig. 3 and presented all regression statistics on individual plots to clarify inputs and significant differences. To prevent confusion, we removed the standard error bars from the plot means and depict 95% confidence intervals around the regressions (74 plots). We also indicate in the caption that both the means and the confidence intervals are at the 74-plot level.

(Lines 321-323)

(Comment) Second, I have a question about the calculations. both the H_c and H_i are based on calculations, and not measured in real life. What conclusions can be drawn from the H_i values, if these are calculations, and do you also have information from some sites based on actual nutrient cycling data? Herbivory now in the SEM leads to H_c and H_i , but because this is based on calculations, with the same data, in particular the H_i link is questionable I think.

(Response) We did not directly measure H_c and H_i in this study. Instead, H_c (gross insect-mediated element flux) was calculated by direct measurements of foliar nutrient concentration, herbivory, and litterfall (to estimate foliar production) from all 74 plots in this study. H_i represents the additional input of elements to the soil released by insects consuming green leaves prior to resorption and was calculated by estimating the flux of elements with and without herbivory. Resorption efficiency and litterfall were also directly measured from every plot to derive estimates of H_i . It is true that the biotic precursors are not fundamentally independent of H_i and H_c since they are included in the calculations used to derive H_c and H_i . The purpose of including them in the SEM is to test H3 – the relative amount and direction of effects that each biotic precursor exerts on the estimates of H_c and H_i . We now clarify this in the text and captions (Lines 168-170, 187-188).

We further describe the distinctions between H_c and H_i in the introduction with additional text in the methods to motivate the importance of this distinction (Lines 188-189, 561-569). “ H_c quantifies the total amount of foliar elements consumed and ultimately released to the ecosystems by herbivores. However, to some extent these elements would have been released anyway to the ecosystem when the leaves abscise. If we assume that leaves abscise at the end of their lifetime after resorption has occurred, and that the vast majority of herbivory occurs before this resorption²⁶, then H_i is designed to calculate the foliar elements released by herbivores that would not otherwise have been released via litterfall. For this reason, H_i is more closely comparable to truly external inputs of relatively labile, mineral nutrients from atmospheric deposition, biological fixation and bedrock weathering.” (Lines 557-565)

(Comment) Third, leaf N is measured after drying at 70%, but it is well described that the N concentration is reduced when drying at such high temperature, and that N is usually measured after drying at lower temperatures, max 60 C i believe.

(Response) Although drying at 60 degrees C is also a reasonable option, drying plant material for nutrient analysis at 70 degrees C is a well-cited and standard method in plant nutrient studies:

Wolf B. A comprehensive system of leaf analyses and its use for diagnosing crop nutrient status. *Communications in Soil Science and Plant Analysis*. 1982 Jan 1; 13(12):1035-59.

Litton CM, Giardina CP, Selmants PC, Sparks JP. Impact of mean annual temperature on nutrient availability in a tropical montane wet forest. *Frontiers in Plant Science*. 2020 Jun 12; 11:500586.

(Comment) Line 185 says: we anticipate that soil nutrient concentration (SE) remains unchanged with H , even as element cycling increases. I wonder if this is true. If i understand it correctly, nutrient cycling is thought to be affected but it does not change the actual concentration in the soil, ie all the nutrients are taken up? Why is this true and why is this important. Please explain.

(Response) Given that this is not tested in our study and represents a small part of the figure, we have chosen to remove the sentence, “While not tested here, we anticipate that soil nutrient concentration (SE) remains unchanged with H , even as element cycling increases.” (Lines 189-190)

(Comment) Minor comment: values in Appendix D are not presented in percentages (e.g. for leaf N) even though it says so in the table

(Response) Thanks to the reviewer for pointing out this error. We have converted proportions to percentages in Appendix D. (Line 870)

REVIEWERS' COMMENTS

Reviewer #1 (Remarks to the Author):

Lines 147-152 The information for the time and frequency of measurement should be given. Did you measure one time in a particular year in each plot? Or, did you measure the variables several times in a whole year, and then quantify the annual fluxes?

Line 241-247 I challenge the assertion here. While the insect mediated nutrient flux is quantitatively larger than nutrient deposition in some areas, that does not mean the insect-mediated nutrient would be important for the nutrient balance or soil availability in those areas. For any ecosystem, nutrient deposition represents the external nutrient sources. In contrast, insect-mediated nutrient is the internal nutrient, without any external inputs for the ecosystem. All (or most of) the nutrients mediated by insects exist in the ecosystem, even if there is no insect in the ecosystem. Insects do not provide any new nutrient for the ecosystem. If there is no insect, such amount of nutrients would be present in other parts of the ecosystem, such as the litters.

Lines 261-263 This case study from European Russia was based on boreal forest, which is quite different from broadleaved forest. Thus, the lower annual loss rate led by insect outbreak in boreal forest does not provide strong evidence for the importance of background levels of insect herbivory in broadleaved forests.

Line 292 the comma should be moved after the ()

Line 315 Using 'P' instead of 'PValue' in all the figures

Line 488 'leaf litter' instead of 'litter leaf'

Line 548 Authors have reported the data of C resorption. The method used to calculate carbon resorption should be added.

Reviewer #2 (Remarks to the Author):

I have read the revision and the questions i had have all been answered. I have no further questions. As stated before, i find this an interesting and important study and recommend publication in nature communications. The topic addressed and the insights merit publication in this top journal in my opinion.

[**editorial note:** reviewer name redacted]

We thank the reviewers for continued support of our manuscript and additional comments. We have responded to new comments from the reviewers below.

REVIEWERS' COMMENTS

Reviewer #1 (Remarks to the Author):

(Comment) Lines 147-152 The information for the time and frequency of measurement should be given. Did you measure one time in a particular year in each plot? Or, did you measure the variables several times in a whole year, and then quantify the annual fluxes?

(Response) Thank you for pointing out this need for clarification in the introduction. We have tried to clarify our methods without being verbose in the following way: “We used standardized methods to *regularly* collect and analyze green and freshly senesced leaves *throughout the growing season for one or two years at each plot* for: (i) foliar biomass production, (ii) foliar herbivory, and (iii) foliar C, N, P and Si concentrations. We then used these measures to calculate *annual* stand-level element fluxes generated during leaf consumption by entire, natural communities of insect herbivores under non-outbreak conditions¹⁰.” (Lines 150-155)

(Comment) Line 241-247 I challenge the assertion here. While the insect mediated nutrient flux is quantitatively larger than nutrient deposition in some areas, that does not mean the insect-mediated nutrient would be important for the nutrient balance or soil availability in those areas. For any ecosystem, nutrient deposition represents the external nutrient sources. In contrast, insect-mediated nutrient is the internal nutrient, without any external inputs for the ecosystem. All (or most of) the nutrients mediated by insects exist in the ecosystem, even if there is no insect in the ecosystem. Insects do not provide any new nutrient for the ecosystem. If there is no insect, such amount of nutrients would be present in other parts of the ecosystem, such as the litters.

(Response) The statement here refers to H1 which makes the distinction not between internal or external fluxes, but between labile and recalcitrant fluxes (Lines 163-165). We have now carefully screened the text throughout to ensure that this comparison is clear (Lines 133-138, 155-156, 234-236, 283-285, 520-522, 572-577). We believe this is a more ecologically relevant comparison because there is ample ecological evidence that most ecosystem nutrients occur in relatively inaccessible organic forms, and that ecosystems respond not to overall nutrient abundance, but to the abundance of labile, plant/microbe accessible nutrients (Walker & Syers, 1976; Vitousek & Reiners, 1975; Reiners et al., 1981; Menge et al., 2012). So, while it is true that the presence of insects will not alter the total quantity of nutrients in a system (soil and plants), this metric is poorly related to ecosystem processes. Instead, we argue (Lines 130-140, 178-193) that the important effects of herbivorous insects derive from altering pathways of nutrients within the system and the relative proportion of labile versus recalcitrant nutrients. There is ample evidence that herbivores produce more labile material than the leaves they consume (Bardgett & Wardle, 2010; Hartley & Jones, 2008; Frost & Hunter, 2004; Kaukonen et al., 2013). We thus believe that the more appropriate fluxes to compare herbivore mediated fluxes with are other labile fluxes such as deposition and weathering.

Furthermore, insect herbivory can impact host-specific nutrient conservation by transferring foliar nutrients to soil before translocation mechanisms can conserve nutrients for the host species. This shortcutting hurts host species while making a large pool of nutrients available to other species, especially beneficial to species not impacted by herbivory (Bardgett et al., 1999). This redistribution of nutrients might either increase diversity if host species are dominant (i.e., herbivory enhances nutrient status of less common species) or reduce diversity if impacted species are rare. Insect herbivores thus have the capacity to influence plant community composition, which in turn affects decomposer activity and consequently the supply of nutrients to plants from the soil (Hartley & Jones, 2008; Bardgett & Wardle, 2010).

Bardgett, R. D., Cook, R., Yeates G. W., Denton, C. S. The influence of nematodes on below-ground processes in grassland ecosystems. *Plant Soil* **212**, 23-33 (1999).

Bardgett, R. D. & Wardle, D. A. Aboveground–belowground Linkages: Biotic Interactions, Ecosystem Processes, and Global Change (Oxford University Press, Oxford, 2010).

Frost, C. J. & Hunter, M. D., Insect canopy herbivory and frass deposition affect soil nutrient dynamics and export in oak mesocosms. *Ecology*, **85**, 3335-3347 (2004).

Hartley, S. E. & Jones, T. H. Insect Herbivores, Nutrient Cycling and Plant Productivity. In *Insects and Ecosystem Function* Ch. 2 (Springer–Verlag, Heidelberg, 2008).

Kaukonen, M., Ruotsalainen, A. L., Wäli, P. R., Männistö, M. K., Setälä, H., Saravesi, K., Huusko, K. & Markkola, A. Moth herbivory enhances resource turnover in subarctic mountain birch forests? *Ecology*, **94**, 267-272 (2013).

Menge, D. N., Hedin L. O., Pacala, S. W. Nitrogen and phosphorus limitation over long-term ecosystem development in terrestrial ecosystems. *PLoS One* **7**, e42045 (2012).

Reiners, W. A. Nitrogen cycling in relation to ecosystem succession. *Ecol. Bull.*, **33**, 507-528 (1981).

Vitousek, P. M. & Reiners, W. A. Ecosystem succession and nutrient retention: a hypothesis. *BioSci*, **25**, 376-56 (1975).

Walker, T. W., & Syers, J. K. The fate of phosphorus during pedogenesis. *Geoderma*, **15**, 1-19 (1976).

(Comment) Lines 261-263 This case study from European Russia was based on boreal forest, which is quite different from broadleaved forest. Thus, the lower annual loss rate led by insect outbreak in boreal forest does not provide strong evidence for the importance of background levels of insect herbivory in broadleaved forests.

(Response) We thank the reviewer's evaluation of this statement. We have decided to remove these lines from the manuscript because the preceding text and references are sufficient to make the point.

(Comment) Line 292 the comma should be moved after the ()

(Response) Thank you for pointing out this typo. We have removed the comma after the parentheses. (Line 304)

(Comment) Line 315 Using 'P' instead of 'PValue' in all the figures

(Response) We have replaced “PValue” with “*P*” in the micrographs. (Line 326)

(Comment) Line 488 ‘leaf litter’ instead of ‘litter leaf’

(Response) We have replaced “litter leaf” with “leaf litter” in the subheading. (Line 497)

(Comment) Line 548 Authors have reported the data of C resorption. The method used to calculate carbon resorption should be added.

(Response) Thank you for noticing this oversight. We have added methods of calculating C resorption in the text alongside existing methods of calculating N and P resorption. (Lines 559-569)

Reviewer #2 (Remarks to the Author):

(Comment) I have read the revision and the questions i had have all been answered. I have no further questions. As stated before, i find this an interesting and important study and recommend publication in nature communications. The topic addressed and the insights merit publication in this top journal in my opinion.

(Response) We thank the reviewer for continued support of the study.